

PeerJ Hubs
Published on behalf of

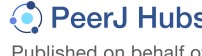
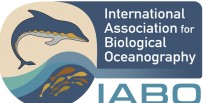

# Biology and ecology of the lionfish *Pterois volitans/Pterois miles* as invasive alien species: a review

Laura del Río[1],[*], Zenaida María Navarro-Martínez[1],[*], Dorka Cobián-Rojas[2], Pedro Pablo Chevalier-Monteagudo[3], Jorge A. Angulo-Valdes[4] and Leandro Rodriguez-Viera[5]

[1] Center for Marine Research, Universidad de La Habana, La Habana, Cuba
[2] Guanahacabibes National Park, Center of Research and Environmental Services, Ministry of Science, Technology and Environment, Pinar de Río, Cuba
[3] National Aquarium of Cuba, La Habana, Cuba
[4] Eckerd College, St. Petersburg, FL, United States of America
[5] Faculty of Marine and Environmental Sciences, Campus de Excelencia Internacional del Mar (CEIMAR), University of Cadiz, Puerto Real, Cadiz, Spain
[*] These authors contributed equally to this work.

Corresponding author
Leandro Rodriguez-Viera,
leokarma@gmail.com

## ABSTRACT

The lionfish is an exotic invasive fish native to the Indo-Pacific, which is established in the western Atlantic Ocean and the Caribbean Sea. Lionfish can affect native fishes and invertebrates through direct predation or competition for food. The present review aims to analyze the most relevant characteristics of the biology and ecology of lionfish as an invasive alien species, with an emphasis on Cuba. We provide a current view of the well-known lionfish as a successful invasive fish, and we put in this context the information regarding lionfish in Cuban waters, enriching the background knowledge, and giving novel and relevant information. The compilation of numerous publications on the subject has allowed for a more complete analysis of essential aspects of this invader in the Cuban archipelago. The consulted literature records that the first report of lionfish in Cuba occurred in 2007; subsequently, sightings of lionfish were reported in numerous localities. In 2010, the lionfish was considered an invasive alien species, which currently is established in various habitats, at depths up to 188 m, throughout the Cuban archipelago (*e.g.*, coral reefs, mangroves, seagrass beds, submerged artificial structures). In addition, it has reached very high densities (12.42 ind./100 m$^2$), which exceed those reported in the Indo-Pacific as well as in many locations in the Western Atlantic. It has been confirmed that the lionfish in Cuba also presents numerous characteristics that guarantee its success as an invader, among them: less quantity and diversity of parasites than other Atlantic fishes found in similar environments, a high number of gametes in the gonads, reproductive activity during all year and wide diet. The most important fish families for the lionfish diet in Cuba have been Pomacentridae, Gobiidae, Scaridae, Holocentridae, Mullidae, Labridae and Acanthuridae; and the most important crustacean orders are Decapoda, Mysida, Stomatopoda and Isopoda. In Cuba, as in the entire invaded region, numerous investigations have been directed to evaluate the impact of this invader on ecosystems, and although there is enough information, their results differ. Additional studies are required to assess the impact of lionfish as a predator after several years of invasion on a larger geographic scale in Cuba and

other areas of the region. This knowledge will allow the development of more effective control strategies. Periodic lionfish culling have been carried out in Cuban MPAs as a control strategy, and some positive results have been observed, such as the average size reduction; however, further efforts are still required. Due to the importance of the study of lionfish as an invader, this review is a necessity as it provides, for the first time, a comprehensive analysis of lionfish information and results from Cuba, which is adequately contrasted with previous studies of other areas, particularly, from the Greater Caribbean.

## INTRODUCTION

The lionfish (*Pterois volitans* (Linnaeus, 1758)/*Pterois miles* (Bennett, 1828) complex) is considered an invasive alien species (IAS), because it occurs in territories outside its native region and constitutes a threat to biodiversity in the invaded area. IAS can cause substantial changes in recipient ecosystems (*Blackburn et al., 2014*) and are among the main risks associated with the extinction of various organisms (*Bellard, Cassey & Blackburn, 2016*). For these reasons, its study, management, and control should be considered a priority.

Native to the Indo-Pacific, lionfish is established successfully in the western Atlantic Ocean and the Caribbean Sea as an invader (*Morris & Akins, 2009*; *Schofield, 2010*). The first record in the invaded area dates from 1985, in Florida, the United States of America (USA) (*Morris & Akins, 2009*). During the following years, it showed up along the coasts of several countries (*Schofield, 2010*). The first report of lionfish in Cuba occurred in 2007; subsequently, observations of lionfish were reported in numerous localities of the country (*Chevalier-Monteagudo et al., 2008*). In 2010, sightings and capture reports already indicated its presence in the USA southeast coast, most of the Caribbean, the Gulf of Mexico and some areas of South America (*Schofield, 2010*).

*P. volitans* and *P. miles* have very similar morphological characters in the invaded region; therefore, the only way to accurately identify them in their non-native distribution range is through molecular analysis (*Hamner, Freshwater & Whitfield, 2007*; *Morris & Whitfield, 2009*; *Burford Reiskind et al., 2019*). The use of the *P. volitans/P. miles* complex or simply "lionfish" is preferred to refer to the two species present in the Western Atlantic and the Caribbean Sea, since in the absence of a molecular analysis it is not possible to be certain about which of the individuals analyzed belong to each species.

Although these species have successfully invaded estuaries, seagrass beds and mangroves, coral reefs constitute one of the most affected ecosystems (*Morris & Akins, 2009*; *Kulbicki et al., 2012*). Coral reefs are subject to numerous stressful factors (*Bruno, Côté & Toth, 2019*) and could suffer an accelerated deterioration with this invader pressure. For instance, lionfish could harm reef health by affecting native fishes and invertebrates through direct predation (*Green & Côté, 2010*; *Cobián-Rojas et al., 2018b*) and competition for food

(*Morris & Akins, 2009*; *Albins, 2013*). In addition to possible ecological damage, it could also cause serious economic losses in the invaded area, by affecting fishing and tourism (*Morris & Whitfield, 2009*; *Morris & Green, 2013*).

The rapid lionfish spread and its potential impact on the invaded ecosystems have made it necessary to carry out new studies that expand the current knowledge about this fish. Valuable reviews have been made that address several topics such as invasion, impacts, control, reproduction, life history, phylogeography and genetics (*Côté, Green & Hixon, 2013*; *Rittermann, 2016*; *Côté & Smith, 2018*; *Díaz-Ferguson & Hunter, 2019*). However, in these reviews information about lionfish in Cuba is absent or very scarce, despite the relevance of studying lionfish in this country, and the existence of numerous Cuban researchers related to the subject. Knowledge of lionfish in Cuba is essential for the comprehensive management of these IAS in the Western Atlantic and the Caribbean Sea, since its geographical location makes it a key country within the invaded area. For instance, Cuba may be the largest exporter of lionfish larvae with high levels of connectivity to all the other precincts (*Johnston & Purkis, 2015*). The present review aims to analyze the most relevant characteristics of the biology and ecology of lionfish as IAS, with an emphasis on Cuba. It provides, for the first time, a comprehensive analysis of lionfish information and results from Cuba, which is adequately contrasted with previous studies of other invaded areas, particularly, from the Greater Caribbean. The information contained in this review may be of great use to researchers in the field of lionfish biology and ecology in the invaded area, and provides greater visibility into the history, characteristics, and impact of this invader in the largest of the Antilles.

## SEARCH METHODOLOGY

This review is not focused on analyzing all the lionfish scientific knowledge, rather is focused on the comprehensive analysis of the most important aspects of lionfish and the main gaps in current knowledge and management in this field. It was prepared during five years of an exhaustive search of published articles related to the biology and ecology of the lionfish, and other aspects of interest that may contribute to understanding its natural history and management. Google Scholar and ScienceDirect were used with different search terms (*e.g.*, "lionfish", "lionfish invasion", "lionfish biology and ecology"). Alternative keywords were also used, *e.g.*, "*Pterois volitans*", "*Pterois*", combined with terms related to geographic locations: "Caribbean", "invaded areas", and "native area". Although research articles on lionfish in its native distribution range were also reviewed, priority was given to papers on lionfish, as an invader, in the Western Atlantic and the Caribbean Sea. Particular relevance was given to Cuban studies, which also include graduate thesis. Some articles, not directly dealing with lionfish, were included since they provide useful information for the review. Articles in English and Spanish language were taken into account, published from 1985 (the first report of lionfish in the invaded region) until 2022; except for a key article from 1959 focused on the lionfish venom as the main defense mechanism in these species.

More than 170 articles were consulted, of which more than 150 contributed to the article main text. In addition, the native distributions of *Pterois volitans* and *Pterois miles*
were added as Supplementary Material, which is based on the information available in FishBase (AquaMaps: *Kaschner et al., 2019*). With the information from 18 diet studies from different invaded regions, a graph represents the most common fish families and all the crustaceans, mollusks, and algae/seagrass identified on lionfish and the number of studies in which they were observed. The graph was created using the R language (*R Core Team, 2020*) and the ggplot2 package (*Wickham, 2016*). A total of 91 research articles were used to summarize the current knowledge about lionfish. These articles were classified into nine themes: Abundance, Toxicology, Control, Diet, Genetic/Phylogenetic, Impact, Invasion, Reproduction/Development and Behavior. Using this later information and the geographic coordinates of each study location, the map with the distribution of studies by subject was produced. The program QGIS, version 3.4.5-Madeira, was used for all the map creation.

# NATURAL AND INVADED GEOGRAPHICAL DISTRIBUTION ZONES

The natural distribution of *P. volitans* includes the Indian and Pacific oceans, and covers a very large area; while *P. miles* is native to the Red Sea, the Persian Gulf and the Indian Ocean, except for Western Australia (Fig. S1) (*Schultz, 1986*). The first sighting of lionfish in the Western Atlantic occurred in Florida in 1985 (*Morris & Akins, 2009*). Other observations were recorded subsequently in 1992 (Courtenay, 1995 cited by *Morris & Akins, 2009*). During the following years, sporadic appearances of the species occurred along the USA coasts (*Whitfield et al., 2002*; *Schofield, 2009*).

Lionfish were reported in the Bermudan Islands in 2000 (*Whitfield et al., 2002*), in the Bahamas in 2004, in the Turks and Caicos Islands in 2006, in Colombia, the Cayman Islands, Jamaica, Puerto Rico, Haiti, Belize, and the Dominican Republic in 2008 (*Guerrero & Franco, 2008*; *Schofield, 2009*). In 2009 it was registered in Mexico, Panama, Honduras, Costa Rica (*Schofield, 2009*), and Nicaragua (*Schofield, 2010*); and in 2014 in Brazil (*Ferreira et al., 2015*).

*Chevalier-Monteagudo et al. (2008)* made the first report of lionfish in Cuba in 2007, from the capture of two specimens of *P. volitans*, in Los Caimanes Keys southern coast, Villa Clara. Subsequently, observations of *P. volitans/P. miles* were reported in numerous localities of the country (*Chevalier-Monteagudo et al., 2008*). It is estimated an initial settlement on the northern coast between late 2006 and early 2007, and from 2009, when they colonized the southern coast from east to west (*Chevalier-Monteagudo et al., 2013a*). Lionfish populations were established in 2011 on the northern coast and later in the southern coast (*Chevalier-Monteagudo et al., 2013a*). In 2010, the lionfish was considered an IAS established throughout the Cuban archipelago.

## Density in its distribution areas

Along its natural distribution areas, ca. 0.25−0.26 ind./100 m$^2$ (ind.: lionfish) average densities have been recorded, with a range from 0 to 1.11 ind./100 m$^2$ (*Darling et al., 2011*; *Kulbicki et al., 2012*) (Table 1). On the contrary, in the Atlantic Ocean, its density tends to be higher, *e.g.*, >1 ind./100 m$^2$ in the Mesoamerican Reef, in Cuba (Jardines

de la Reina) (*Hackerott et al., 2013*), and in the Bahamas (*Green & Côté, 2009*; *Darling et al., 2011*) (Table 1). Even in ichthyoplankton, lionfish larvae can reach density values comparable to those of common reef fishes (*e.g.*, 0.4–0.7 lionfish larvae/1000 m$^3$ in the Straits of Florida) (*Sponaugle et al., 2019*). In Cuba, reported lionfish densities are relatively high (2.1 ind./100 m$^2$) (*Chevalier-Monteagudo, 2017*), compared to those documented in the Indo-Pacific (*Darling et al., 2011*; *Kulbicki et al., 2012*) and in other locations in the Western Atlantic (Table 1). In the Guanahacabibes National Park (GNP) it is considered one of the most abundant species on coral reefs and densities above 5.0 ind./100 m$^2$ have been recorded (*Cobián-Rojas et al., 2016*; *Cobián-Rojas et al., 2018a*); while 12.42 ind./100 m$^2$ have been found in the tourist coast of Holguín (*Reynaldo et al., 2018*). One of the main concerns in Cuba is the ability of lionfish populations to grow rapidly in numbers. On the coast of Holguín, in just one year, there was a notable increase in the density of this invader (1.01 ind./100 m$^2$ in 2013, 7.5 ind./100 m$^2$ in 2014) (*Vega et al., 2015*; *Reynaldo et al., 2018*), a fact that shows the need to strengthen control strategies.

### New invasions

Lionfish have invaded also successfully the Mediterranean Sea. In 1991, *P. miles* was reported for the first time in this region, after the capture of one individual in Herzliya, Israel (*Golani & Sonin, 1992*). For more than 20 years, no other reports of lionfish occurred in the area, suggesting that the species had not yet become established (*Bariche, Torres & Azzurro, 2013*; *Kletou, Hall-Spencer & Kleitou, 2016*; *Al Mabruk et al., 2020*). But, in 2012 two individuals of the species *P. miles* were captured, and subsequently, several reports of lionfish occurred, which evidenced their establishment in the Mediterranean Sea (*Kletou, Hall-Spencer & Kleitou, 2016*; *Azzurro et al., 2017*; *Al Mabruk & Rizgalla, 2019*; *Al Mabruk et al., 2020*). In this newly invaded area lionfish have been observed mainly near rocks, but also on sandy bottoms and artificial structures; generally, in waters from 4 to 42 m deep, with a peak between 20 and 29 m (*Azzurro et al., 2017*).

Some authors consider the Suez Canal as a very probable route of introduction (*Bariche, Torres & Azzurro, 2013*; *Bariche et al., 2017*; *Dimitriou et al., 2019*). There is evidence that the eastern Mediterranean population of *P. miles* may have originated from the Red Sea population (*Bariche et al., 2017*; *Stern et al., 2018*). Further geographical expansion in the Mediterranean Sea could occur in the coming years. *Poursanidis et al. (2020)* found evidence that *P. miles* could reach the Strait of Gibraltar within a few years.

## FACTORS THAT HAVE ALLOWED ITS SUCCESSFUL ESTABLISHMENT

The speed and success of the lionfish expansion in the Atlantic Ocean and the Caribbean Sea indicate that it has been favored by its biological and ecological traits, as revealed by several studies. For instance, it has characteristics that are predictors of the ability of non-native fishes to invade territories outside its natural range (*Morris & Whitfield, 2009*; *Côté & Smith, 2018*): extensive native distribution range (*Schultz, 1986*), wide diet (*Morris & Akins, 2009*; *Cure et al., 2012*), ability to tolerate long periods of fasting (up to 10 weeks of fasting) (*Fishelson, 1997*), traits that allow consuming large and elusive prey (stomach

**Table 1 Lionfish density recorded at different locations in its natural range and invaded area (density in individuals/100 m².).**

| Locality | Year | Density[*] | Species | Reference |
|---|---|---|---|---|
| Native | | | | |
| Eilat, Israel (Red Sea) | | ~0.80 | *P. volitans* | *Fishelson (1997)* |
| Kenya | 2010 | 0.25 ± 0.46 SD (0–1.11) | *P. miles* | *Darling et al. (2011)* |
| Invaded (Western Atlantic and Caribbean Sea) | | | | |
| North Carolina, USA | 2004 | 0.21 ± 0.51 SD (0–0.73) | *Pterois* spp. | *Whitfield et al. (2007)* |
| Bahamas | 2008 | 3.93 ± 1.44 SD | *P.volitans* | *Green & Côté (2009)* |
| Bahamas | 2008 | 1.00 ± 1.03 SD | *P. volitans* | *Darling et al. (2011)* |
| Costa Rica | 2011 | 0.92 ± 1.29 SE | *Pterois* spp. | *Sandel (2011)* |
| Jardines de la Reina National Park, Cuba | 2009-2012 | 1.50 | *Pterois* spp. | *Hackerott et al. (2013)* |
| Mesoamerican Barrier Reef, Belice and Mexico | 2009-2012 | 1.60 | | |
| Curaçao (unfished locations) | 2011 | 1.27 | *Pterois* spp. | *De León et al. (2013)* |
| Bonaire (fished locations) | | 0.30 | | |
| Bonaire (unfished locations) | | 0.70 | | |
| Bahamas | 2009 | 0.13 ± 0.18 SD (0–0.73) | *P. volitans* | *Anton, Simpson & Vu (2014)* |
| Venezuela | | 0.30 | *P. volitans* | *Agudo & Klein Salas (2014)* |
| Northern Gulf of Mexico (natural reefs) | 2013 | 0.49 (0–1.8) | *P. volitans* | *Dahl & Patterson III (2014)* |
| Northern Gulf of Mexico (artificial reefs) | | 14.7 (0–38.5) | | |
| Guanahacabibes National Park, Cuba | 2011 | ≥5.0 | *P. volitans* | *Cobián-Rojas et al. (2016)* |
| | 2013 | 3.10 | | |
| Cuba | 2007-2013 | 2.10 (0–5,7) | *P. volitans* | *Chevalier-Monteagudo (2017)* |
| Cayos de San Felipe National Park, Cuba (mangrove) | 2013-2015 | 0.65 | *P. volitans* | *Guardia et al. (2017)* |
| Cayos de San Felipe National Park, Cuba (reefs at 15 m) | | 0.41 | | |
| Cayos de San Felipe National Park, Cuba (reefs at 25 m) | | 0.32 | | |
| Bacalar Chico Marine Reserve, Belize | 2014 | 0.27 ± 0.09 SE (0–2.67) | *Pterois* spp. | *Anderson et al. (2017)* |
| Bermuda | 2013-2015 | 1.96 (0–7.60) | *Pterois* spp. | *Goodbody-Gringley et al. (2019)* |
| Florida shelf, USA (natural reefs) | 2014 | 0.57 | *P. volitans* | *Dahl, Edwards & Patterson III (2019)* |
| | 2015 | 0.34 | | |
| Florida shelf, USA (artificial reefs) | 2014 | 32.98 | | |
| | 2016 | 20.45 | | |
| Veracruz, Mexico | 2018 | 0.004–0.04 | *P. volitans* | *Aguilar-Medrano & Vega-Cendejas (2020)* |
| Invaded (Mediterranean Sea) | | | | |
| Cape Greco Marine Protected Area, Cyprus | 2019 | 0.1 ± 0.06–1.05 ± 0.13 SE | *P. miles* | *Kleitou et al. (2021)* |
| Samandağ coast, Turkey | | 0.024 | *P. miles* | *Turan & Doğdu (2022)* |

**Notes.**

*Density given as mean, or minimum and maximum between parentheses.

SD, Standard deviation; SE, Standard error.

with the ability to expand 30–32 times, the large diameter of its mouth, high suction capacity, large protrusion of the premaxilla) (*Fishelson, 1997*; *Rojas-Vélez, Tavera & Acero, 2019*), and preys that not recognize it as a predator (*Albins & Lyons, 2012*; *Berchtold & Côté, 2020*). In addition, lionfish have: high physical tolerance (*Fishelson, 1997*), effective defense mechanisms [spines loaded with poison] (*Saunders & Taylor, 1959*; *Whitfield et al., 2007*), absence of effective biotic resistance mechanism in the invaded area (*Hackerott et al., 2013*; *Valdivia et al., 2014*; *Cobián-Rojas et al., 2018b*), resistance to ectoparasites (*Côté & Smith, 2018*), rapid growth (e.g., growth rate (K) equivalent to 0.42 for an entire population sampled in Little Cayman, with estimates of 0.38 for males and 0.57 for females (*Edwards, Frazer & Jacoby, 2014*), although values higher than these have been found by other authors (see *Côté & Smith, 2018*)); early sexual maturity (*Bustos-Montes et al., 2020*), long breeding season (all year) (*Morris & Whitfield, 2009*; *Morris, Sullivan & Govoni, 2011*; *Gardner et al., 2015*), high fecundity (1800–41 945 eggs per laying) (*Gardner et al., 2015*), larval period with an adequate duration to achieve a wide dispersal (20–35 days) (*Ahrenholz & Morris, 2010*), ability to travel long distances (1.38 km) and even crossing large areas of sand considered barriers to the movement of most reef fishes (*Tamburello & Côté, 2014*), relatively large body size (45 cm in total length, with sexual dimorphism where the male reaches the largest sizes and presents slower growth) (*Whitfield et al., 2007*; *Morris & Whitfield, 2009*; *Darling et al., 2011*; *Edwards, Frazer & Jacoby, 2014*) and relatively high longevity (ca. 10 years) (*Froese & Pauly, 2019*). A recent research based on a sample of 64 lionfish from Cuba, recorded a high number of gametes in the gonads, semicystic spermatogenesis, and reproductive activity all year (*Cruz-López et al., 2020*). All these are clear advantages for lionfish as species, but particularly as an alien and invasive species. Regarding parasitism, lionfish have been shown less quantity and diversity of parasites than other Atlantic fishes in similar environments (*Côté & Smith, 2018*). The parasitic species of lionfish in Cuba by taxonomic groups is poor; and the infection parameters have very low values and differ markedly from those reported in native hosts that live in the same sites as the lionfish analyzed (*Chevalier-Monteagudo et al., 2013b*). This reinforces how has been possible the lionfish establishment and dispersion so successfully and rapidly in the Caribbean and the Western Atlantic.

## HABITATS INVADED BY LIONFISH

Lionfish have occupied a wide range of habitats, including estuaries, seagrass beds, coral reefs, hard and rocky bottoms, channels, and mangroves. It has been observed taking refuge in association with biotic and abiotic structures present in these habitats, *e.g.*, rocks, coral heads, artificial structures, and softer structures such as sponges, walls of blowouts in seagrass beds, mangrove roots, also, in docks for small boats and soft corals (*Claydon, Calosso & Traiger, 2012*; *Pimiento et al., 2013*). The depth range reported for this invader is 0–55 m (*Froese & Pauly, 2019*); however, it has been repeatedly described in mesophotic reefs more than 60 m deep, in the native and invaded areas (*Andradi-Brown et al., 2017*; *Luiz et al., 2021*). There are records of its presence at more than 245 m depth in Curaçao (247 m), Honduras (250 m), the Bahamas (300 m), and Bermuda (304 m) (*Andradi-Brown, 2019*) and reference therein.

In Cuba, the lionfish has been observed in shallow coral reefs (*Chevalier-Monteagudo, 2017*; *Cobián-Rojas et al., 2018a*; *Cobián-Rojas et al., 2018b*) and mesophotic reefs at a maximum depth of 188 m (*Reed et al., 2018*; *Cobián-Rojas et al., 2021*). Additionally, this invader has been recorded in mangroves (*Pina-Amargós, Salvat-Torres & López-Fernández, 2012*; *Guardia et al., 2017*; *Rodríguez-Viera et al., 2018*), seagrass beds (*Del Río et al., 2022*), submerged artificial structures and muddy bottoms with stones (*Chevalier-Monteagudo et al., 2013b*), and it is also usually abundant in artificial shelters used in the lobster fishery (*Rodríguez-Viera et al., 2018*). In Holguín, higher densities of lionfish have been recorded in the fore reefs (0.3 ind./100 m$^2$) compared to other habitats such as seagrass beds (0.1 ind./100 m$^2$) (*Vega et al., 2015*). A comparative study between mangroves and coral reefs in Cayos de San Felipe National Park estimated lower lionfish sizes in mangroves (average size: 126 mm) than those estimated in coral reefs (241–258 mm) (*Guardia et al., 2017*). In Punta Francés National Park (PFNP) lionfish caught in seagrass beds had smaller sizes (average length: 212 mm) than those captured in coral reefs (average length: 252 mm) (*Del Río et al., 2022*). This trend has been observed also in Holguín, where larger individuals have been recorded in the coral reef (*Vega et al., 2015*).

## PREDATORY BEHAVIOR AND DIET

The predatory strategy of the lionfish is practically unique among the predatory fishes from the Western Atlantic and the Caribbean. When the lionfish chooses its target, it corners the prey against a rock or confined space using its large pectoral fins, moving close enough with its fins extended, and attacks quickly (*Côté & Maljkovic, 2010*; *Albins & Hixon, 2011*). In addition, it can expel rapid jets of water that cause a disorder in the lateral line system of the prey fishes, and it creates a small current that disorients, causing the prey movement towards the lionfish mouth (*Cure et al., 2012*). Likely, lionfish prefer to grab their prey by the head, which allows them to avoid accidents with the spines and limits the possibilities of escape (*Albins & Lyons, 2012*). Generally, they swallow the whole prey (*Morris & Green, 2013*).

Numerous studies show that the lionfish is a generalist carnivore capable of varying the composition of its diet depending on the most available preys in the region where it lives (*Arredondo-Chávez et al., 2016*; *Peake et al., 2018*; *Sancho et al., 2018*). However, some studies have shown that lionfish can also exhibit preferences for certain preys (*Chappell & Smith, 2016*; *Ritger et al., 2020*; *Santamaria, Locascio & Greenan, 2020*). After conducting numerous samplings aimed at studying lionfish diet in Cuban locations, a great diversity of food entities was recorded (*Chevalier-Monteagudo et al., 2013b*; *Del Río et al., 2022*). It was observed a strong correlation between the number of entities found in the lionfish diet and the densities in the wild, which includes several fish families and species. This trend suggests that lionfish in these locations feed on the most available prey.

### Diet components

Lionfish have a primarily piscivorous diet, supplemented with crustaceans and other invertebrates such as mollusks and echinoderms (*Green, Akins & Côté, 2011*; *Chevalier-Monteagudo, 2017*; *Dahl et al., 2017*). For instance, numerous fish families have been

recorded in the stomach contents of lionfish (Fig. 1). Crustaceans may also become important in the lionfish diet (*Dahl & Patterson III, 2014*; *Villaseñor Derbez & Herrera-Perez, 2014*). The proportional importance of crustaceans in the diet is inversely related to the size of this predator. Larger lionfish tend to increase fish consumption and decrease crustacean consumption, in addition to preferring larger prey; this ontogenetic change has been observed in numerous studies (*Morris & Akins, 2009*; *Muñoz, Currin & Whitfield, 2011*; *Arredondo-Chávez et al., 2016*; *Peake et al., 2018*; *Sancho et al., 2018*). In Cuba, several studies have been developed aimed at knowing the main components of the lionfish diet (*García, 2015*; *Vega et al., 2015*; *Pantoja, 2016*; *Cobián-Rojas et al., 2016*; *Pantoja et al., 2017*; *Chevalier-Monteagudo, 2017*; *Del Río et al., 2022*). These studies have analyzed the stomach content of lionfish from different Cuban locations. The identified food entities are included in three main groups: fishes, crustaceans and mollusks, in that order of abundance. The most important fish families for the diet have been Pomacentridae, Gobiidae, Scaridae, Holocentridae, Mullidae, Labridae and Acanthuridae. In the case of crustaceans, the orders Decapoda, Mysida, Stomatopoda, and Isopoda, the infraorder Brachyura and shrimps (infraorders Stenopodidea and Caridea, and the superfamily Penaeoidea) have predominated.

## IMPACT OF LIONFISH IN THE INVADED AREA

One of the great concerns that have arisen after the lionfish invasion is its effect on other native organisms since its arrival breaks the existing balance in the invaded ecosystem. Once the invasion has occurred, there is a new predator, a new competitor, and perhaps a new prey, which tends to reach high densities and has characteristics that guarantee its success as an invader (Fig. 2).

### Impact by predation

Numerous studies have shown that the lionfish is capable of producing great effects on coral reef communities through the direct predation of native fishes and invertebrates (*Albins & Hixon, 2008*; *Morris & Akins, 2009*; *Valdez-Moreno et al., 2012*; *Cobián-Rojas et al., 2018b*). Some alarming results show that there is the possibility that it could even cause, at least locally, the extinction of endemic native species of limited distribution (*Côté & Smith, 2018*). As a generalist predator, lionfish includes numerous fishes and invertebrate's species from the Western Atlantic and the Caribbean (*Morris & Akins, 2009*; *Dahl et al., 2017*), and is capable of consuming preys on reefs at a rate higher than that which native populations can assimilate (*Green & Côté, 2010*). The deterioration of coral reefs due to the lionfish presence would also affect other organisms that depend on these ecosystems, so the consequences could be very serious. One of the greatest information gaps is its synergistic effect with additional stressing factors already existing in coral reefs (*e.g.*, eutrophication, sedimentation, increase in water temperature, coral bleaching, overfishing, and the effect of the massive arrival of sargassum) (*Fabricius, 2005*; *Rodríguez-Martínez et al., 2019*; *Eakin, Sweatman & Brainard, 2019*). These interactions can enhance the deterioration of these ecosystems.

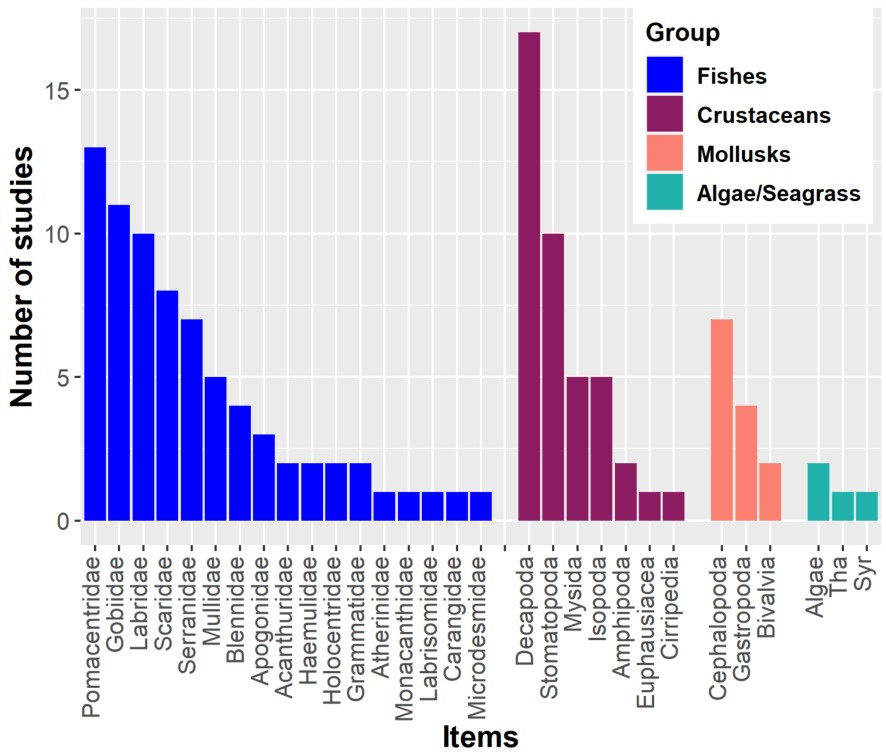

**Figure 1** **Main components of the lionfish diet in the Western Atlantic.** The most common fish families, crustaceans, mollusks and algae/seagrass (Tha. *Thalassia testudinum*, Syr. *Syringodium filiforme*) identified on lionfish diet analyses are showed. Graphic were created with information of diet studies ($n =$ 18) from different invaded regions (*Morris & Akins, 2009*; *Sandel, 2011*; *Muñoz, Currin & Whitfield, 2011*; *McCleery, 2011*; *Valdez-Moreno et al., 2012*; *García, 2015*; *Pantoja, 2016*; *Arredondo-Chávez et al., 2016*; *Cobián-Rojas et al., 2016*; *Hackerott et al., 2013*; *Chevalier-Monteagudo, 2017*; *Pantoja et al., 2017*; *Sancho et al., 2018*; *Peake et al., 2018*; *Aguilar-Medrano & Vega-Cendejas, 2020*; *Santamaria, Locascio & Greenan, 2020*; *Dahl et al., 2017*).

By altering food webs, lionfish could unleash cascading repercussions that affect the entire ecosystem. The possibility of lionfish decimating populations of important herbivores (*i.e.*, parrotfish (Scaridae) and surgeonfish (Acanthuridae)) has been raised, as these have been identified as part of their diet in numerous studies (Fig. 1) (*Albins & Hixon, 2008*; *Morris & Akins, 2009*; *Cure et al., 2012*; *Arredondo-Chávez et al., 2016*; *Chappell & Smith, 2016*; *Chevalier-Monteagudo, 2017*; *Pantoja et al., 2017*). By feeding on herbivorous fishes, the control that these key groups carry out over macroalgae is compromised, which is an essential process for maintaining the coral reefs health (Fig. 2) (*Steneck, Arnold & Mumby, 2014*; *Kindinger & Albins, 2017*). It has been shown that some herbivores such as parrotfish may not recognize the lionfish as a predator, which considerably increases the predation risk (*Berchtold & Côté, 2020*). In addition, lionfish presence may change parrotfish behavior (*e.g.*, decreasing general grazing intensity) and produce an increase of algal cover on coral reefs (*Eaton et al., 2016*; *Kindinger & Albins, 2017*).

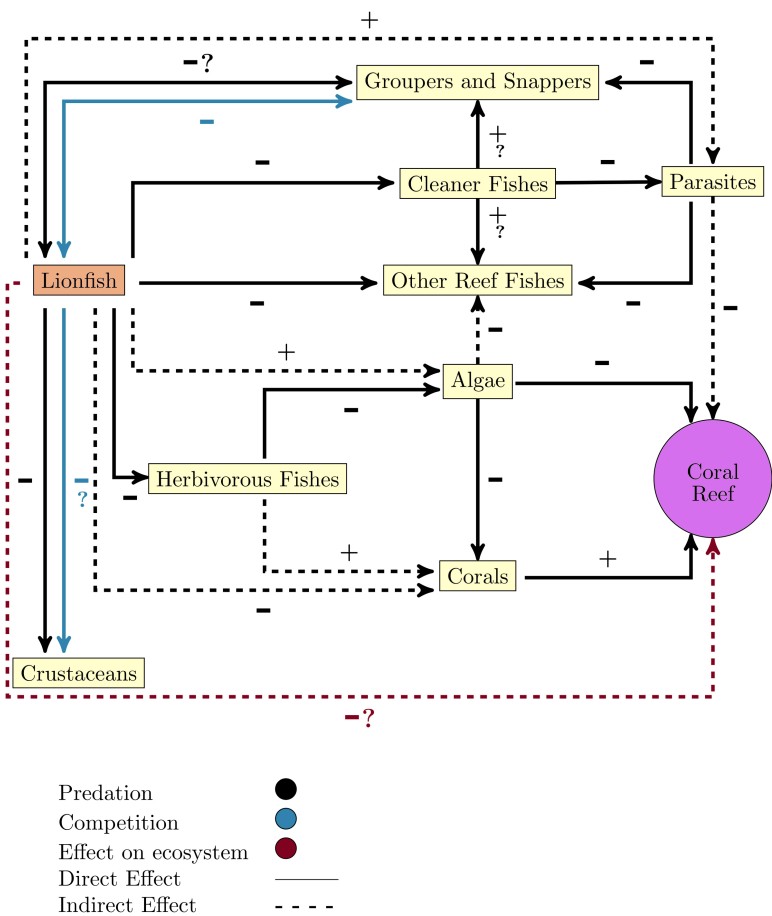

**Figure 2 Diagram of lionfish impact on coral reefs ecosystems and key functional groups.** Note that the line color only apply to relationships between lionfish and the other groups or ecosystem, or if it does not refer to the direct relationship (positive or negative).

Lionfish also can feed on juvenile snappers and groupers (Fig. 2) (*Morris & Akins, 2009*; *Dahl et al., 2017*), two commercially significant groups that can also be potential predators of the lionfish (*Maljkovic & Van Leeuwen, 2008*). Additionally, lionfish feed on cleaning fishes (*e.g.*, *Thalassoma bifasciatum*) which can alter the structure and function of communities on invaded reefs (Fig. 2) (*Tuttle, 2017*).

Some experiments have shown that lionfish can considerably reduce fish recruitment and fish forage, as well as the abundance and richness of native fishes that constitute its prey, an effect that is greater when native predators are also present (*Albins & Hixon, 2008*; *Green et al., 2012*; *Albins, 2013*; *Samhouri & Stier, 2021*). The possibility that these experimental effects can be extrapolated to the entire reef has been contemplated (*Morris & Green, 2013*). However, it should be taken into account that the lionfish effects on native prey in small-scale experiments do not necessarily reflect the results in nature (*Hackerott et al., 2017*). *Hackerott et al. (2017)* studied a larger geographic scale and found no evidence that lionfish appreciably affected the density, richness or composition of prey fishes. In

contrast, a study aimed at examining the impact of lionfish invasion on a wide regional and temporal scale suggested that experimental studies of small-scale reef patches in the Bahamas could provide good indications of the invasion impact in a wider region (*Ballew et al., 2016*).

The lionfish diet studies in different Cuban locations allowed for detecting which are the most affected fishes and invertebrates by direct predation, in addition to providing information about their effects on them (*e.g., García, 2015; Pantoja, 2016; Pantoja et al., 2017*). The analysis of the main ecological relationships of lionfish in Havana reefs showed that the lionfish effect is limited to their most abundant fish prey families and species (*García, 2015*). *Chevalier-Monteagudo (2017)* did not obtain evidence that *P. volitans* had affected the population variables of the usual lionfish preys or the ecosystem ecological variables in several Cuban locations. The new associations detected between native fishes and this invader show the possibility that lionfish are progressively becoming an additional component of the Caribbean's biodiversity (*Chevalier-Monteagudo, 2017*). In addition, no changes were detected in most of the diversity and equity indices of the fish communities in GNP, when comparing several years before and after the lionfish invasion (*Cobián-Rojas et al., 2018a*). These authors discussed that variations in richness, diversity, and equity appear to be more related to reef structure and the effects of fishing than to lionfish abundance. However, in another study conducted in GNP, it was detected that the abundance and size of preys decreased as the abundance of lionfish increased (*Cobián-Rojas et al., 2018b*). Analysis of the density, size, biomass, and diet of lionfish in the GNP suggests that their impact on preys may increase when the invader reaches larger sizes and its populations grow (*Cobián-Rojas et al., 2016*). Additional studies are required to assess the impact of lionfish as a predator after several years of invasion on a larger geographic scale and in different ecosystems invaded in Cuba and in other areas of the region.

## Impact by competition

Lionfish can also affect other organisms through competition for food or shelter (Fig. 2) (*Morris & Whitfield, 2009; Dahl et al., 2017*). This invader occupies the same habitats and consumes the same preys as many native fish species and macroinvertebrates (*Morris & Green, 2013; Arredondo-Chávez et al., 2016; Peake et al., 2018*). A possible competition for food has been evidenced between lionfish and native fishes such as *Cephalopholis cruentata, Cephalopholis fulva, Epinephelus guttatus*, and *Lutjanus apodus*, due to the overlap between their isotopic niches (*Curtis et al., 2017; Eddy et al., 2020; Murillo-Pérez et al., 2021*). Although some researchers have not detected that lionfish affect the density and species richness of their competitors (*Elise et al., 2014; Chevalier-Monteagudo, 2017*), the effect that may have over them, in conditions of low availability of resources, constitutes an important concern. Competition with lionfish could affect the behavior, distribution, growth, survival, and even population size of competitors (*Morris & Green, 2013*). Some studies suggest that lionfish could surpass native predators in the competition for food resources and decrease the abundance of the species that constitute their prey (*Albins & Hixon, 2008; Morris & Akins, 2009; Albins, 2013*). This fact could be explained by taking into account that this invader tends to grow faster (*Albins, 2013; Bustos-Montes et al., 2020*)

and can consume prey at a higher rate than some of these predators (*Albins, 2013*; *Marshak, Heck Jr & Jud, 2018*).

In terms of competition for space, lionfish have a similar habitat preference to *Panulirus argus* and cause an increase in its activity (time spent: active *vs* resting) that could cause a decrease in growth rates and an increase in the risk of predation (*Hunt et al., 2020*). Besides it has been observed that fishes predated by lionfish use shelters similar to those of this invader, which could lead to competition for this resource. However, it has been shown that lionfish share daytime shelters with some of these fishes (*e.g.*, *Gramma loreto*, *Canthigaster rostrata*, *Chromis cyanea*) and the possibility that they act as a client of cleaners or as a protector against other predators is raised (*García-Rivas et al., 2017*).

Cuban researches have been also focused on evaluating the impact of lionfish over its possible competitors. A study performed in six Cuban provinces showed that lionfish densities tend to be similar to or higher than those of their possible competitors (*Chevalier-Monteagudo et al., 2013b*). They take into account the possibility that the invasive fish could displace some native groupers from their ecological niche, and prevent the recovery of their populations. However, no significant correlations were observed between lionfish and the fish families that constitute their potential competitors, in terms of abundance, biomass, and mean size (*Chevalier-Monteagudo et al., 2013b*). *Pantoja et al. (2017)* detected a low overlap between the diets of lionfish and those of the Haemulidae, Serranidae, and Holocentridae families in GNP. It was shown that, although lionfish are essentially piscivorous like snappers (Lutjanidae) and groupers (Serranidae), their diet content differs in the proportion of fishes, crustaceans and mollusks. Therefore, these authors suggest that lionfish probably do not constitute a threat to native fishes of similar trophic levels in the competition for food resources. Additionally, *Chevalier-Monteagudo (2017)* did not detect that *P. volitans* populations altered the population variables of their main competitors in this same location.

## Socio-economic impact

In addition to threatening the ecological functioning and biodiversity of reefs, lionfish represent an economic risk (*Morris & Whitfield, 2009*; *Arredondo-Chávez et al., 2016*). Among the most vulnerable sectors are fishing and tourism, which are of great importance to many countries in the Caribbean and the Atlantic Ocean (*Morris & Green, 2013*). The fisheries sector is affected by the inclusion of juvenile stages of commercially valuable species (*e.g.*, *Epinephelus striatus*, *Ocyurus chrysurus* and *Pagrus pagrus*) in the diet of lionfish (*Morris & Akins, 2009*; *Dahl et al., 2017*). Additionally, the presence of crab and shrimp species important for fisheries (*e.g.*, *Menippe mercenaria* and *Farfantepenaeus duorarum*) has been detected in their diet (*Sancho et al., 2018*). The predatory activity of lionfish on such species of commercial interest could reduce their catches, hinder efforts aimed at the recovery of fishing populations and slow down initiatives aimed at the management and conservation of these groups (*Morris & Akins, 2009*; *Morris & Green, 2013*).

The ecological impact of lionfish can potentially cause a reduction in tourist interest in the most affected areas, due to the decrease in the attractiveness of the reefs. In Cuba the lionfish

has been registered in areas of interest for tourism, *e.g.*, Holguín's tourist coastline (*Vega et al., 2015*; *Reynaldo et al., 2018*), the GNP (*Cobián-Rojas et al., 2016*; *Cobián-Rojas et al., 2018a*; *Cobián-Rojas et al., 2018b*), the PFNP (*Del Río et al., 2022*), Jardines de la Reina National Park (*Pina-Amargós, Salvat-Torres & López-Fernández, 2012*; *Pina-Amargós, Figueredo-Martín & Rossi, 2021*). In such cases, it should be noted that all these represent recreational diving sites, an activity that depends considerably on the attractiveness of the fauna and the ecosystem as a whole. Therefore, the need for adequate control of this invader in these areas is evident, to avoid damaging the ecosystems vital for tourism and thereby affecting the economy. The concern generated in this regard has prompted studies aimed at evaluating the impact of lionfish and the establishment of effective control strategies in some of these affected areas (*Labastida et al., 2015*; *Cobián-Rojas et al., 2016*; *Cobián-Rojas et al., 2018a*; *Cobián-Rojas et al., 2018b*; *Del Río et al., 2022*).

## CONTROL OF LIONFISH POPULATIONS

Since the beginning of the invasion, the potential impact of this IAS has aroused the interest of researchers in detecting possible natural controllers of their populations. In the Caribbean, lionfish have been found in the stomachs of large groupers (*Maljkovic & Van Leeuwen, 2008*; *Mumby, Harborne & Brumbaugh, 2011*; *Côté & Smith, 2018*), which highlights the possibility that groupers can act as a biological controller of this IAS. Strong evidence regarding lionfish predation was obtained pointed to sharks or large groupers as predators since several species have been observed in the area (*Dahl & Patterson III, 2020*).

However, the overfishing of groupers (*Whitfield et al., 2007*) and predation effects during their juvenile stages by lionfish (*Morris & Akins, 2009*; *Villaseñor Derbez & Herrera-Perez, 2014*), may have hindered grouper's effectiveness as natural control in the Caribbean. In subsequent studies carried out in Cuba, Bahamas, Belize, Mexico and Colombia, no evidence of predation by potential predators (*i.e.*, groupers and snappers), has been observed (*Valdivia et al., 2014*; *Cobián-Rojas et al., 2018b*; *Rojas-Vélez, Tavera & Acero, 2019*). Additionally, in a recent study, the stomach contents of more than 200 groupers of five species were analyzed, and the presence of lionfish was not observed among their prey (*Smith & Côté, 2021*). Such results suggest that the invasion of lionfish cannot be controlled by its potential predators, even when the latter can show relatively high biomass, *e.g.*, average biomasses of 7.6 g/m$^2$ in Caribbean reefs (*Valdivia et al., 2014*) and 20.0 g/m$^2$ in Exuma reefs (Bahamas) (*Mumby, Harborne & Brumbaugh, 2011*).

In the absence of effective natural control over lionfish in the Caribbean and Western Atlantic, the development of management plans targeting these species has become a necessity. Lionfish extractions are of great importance as a control strategy, since it has been observed that they can decrease their density, and therefore, the potential impact on native prey fishes and over the most vulnerable components of the ecosystem (*Frazer et al., 2012*; *Côté et al., 2014*; *Harris et al., 2019*; *Samhouri & Stier, 2021*). In addition to density, lionfish sizes have decreased in response to this strategy in various regions within the invaded area, *e.g.*, Bonaire, USA, Bahamas, Cayman Islands, Cuba, and Mexico (*Frazer et al., 2012*; *Akins, 2013*; *Cobián-Rojas et al., 2018a*; *Cobián-Rojas et al., 2018b*). The success

of removals could be significantly increased if coordinated removal programs are carried out in connected areas (*Díaz-Ferguson & Hunter, 2019*).

Cuba is a key exporter of lionfish larvae, so it should be a prime target for lionfish control efforts, and therefore a primary location to implement a comprehensive lionfish culling program (*Johnston & Purkis, 2015*). Periodic lionfish captures have been carried out in Cuban MPAs, as is the case of the GNP and the PFNP. In PFNP, the average size of lionfish tended to decrease over time, evidencing the effectiveness of systematic extractions performed in the area (*Del Río et al., 2022*). For instance, after the largest extraction of lionfish during that study (226 individuals, average length: 259.52 mm), a notable decrease was observed in the average lengths of the specimens (210.63 mm) during the following year, which is attributed precisely to the effect of these catches (*Del Río et al., 2022*). In GNP the International Lionfish Fishing Tournament is held annually, unique of this kind in Cuba. To date, five tournaments have been held, which are focused on controlling lionfish through mass catches by divers from local communities, and national and foreign professional divers; at the time that is promoting the species consumption by local communities. In the framework of these events, 660 lionfish individuals have been caught in the modality of freediving and autonomous diving, with the participation of 103 fishermen (Cobián-Rojas, pers. comm., GNP, Pinar del Río, Cuba). The catches have shown a tendency towards the decline of the species in a period of five years in GNP. However, these massive captures have been made in a small sector of this area (less than 10% of the total area), and although they are also captured in the framework of other activities such as scientific expeditions and guided dives carried out by the International Diving Center María la Gorda, it is considered that the control of this invader is insufficient in the GNP. Far from diminishing the importance of the tournaments, this is an example of systematic control of invasive species and contributes directly to the conservation of the coral reefs in GNP and the region. Importantly, in other Cuban areas (non-marine protected areas), lionfish populations have decreased likely as a result of extractive artisanal fisheries (*Chevalier-Monteagudo et al., 2013b*).

## USES OF LIONFISH

Although lionfish presence is seen as a negative component in invaded ecosystems, they can provide benefits to humans. They can be used as a food source, as an indicator species to detect contamination, in aquarium exhibits, and for obtaining biomedical products (*Sri Balasubashini et al., 2006*; *Chel-Guerrero et al., 2020*; *Van den Hurk et al., 2020*). The use of lionfish as food should be a priority in new studies since it constitutes an alternative for reducing its density in the invaded area. In Cuba, lionfish is recognized as an edible species and is consumed daily in coastal communities (*Chevalier-Monteagudo et al., 2013b*). Educating people about the environmental problems caused by lionfish and its possible use as food can favor a possible market for lionfish meat (*Simnitt et al., 2020*; *Blakeway, Ross & Jones, 2021*). If demand is sufficient, lionfish populations could be reduced to levels that allow the restoration of native ecosystems. In addition, it could contribute to reduce the pressures on native species that are overexploited as fishing resources (*Chapman et*

*al., 2016*). Studies carried out in Cuba, Jamaica and Curaçao showed that lionfish meat is safe (*Hoo Fung et al., 2013*; *Ritger, Curtis & Chen, 2018*; *Squadrone et al., 2020*). Besides, it is valuable as a source of minerals (*e.g.*, potassium and magnesium) and high-quality proteins and peptides, as evidenced by their amino acid composition (*Castro-González et al., 2019*; *Chel-Guerrero et al., 2020*). With 100 g of lionfish fillet, 36–44% of the daily protein requirements for adult people can be supplied (*Castro-González et al., 2019*). Lionfish management as generator of such benefits may be possible and it could represent a profitable and efficient tool in its population control.

## FINAL CONSIDERATIONS

Considering other review articles, which have been focused on the analysis of scientific publications per topic and year, we provide a current view of the well-known lionfish as a marine successful invasive fish. Notably, we provided relevant information regarding lionfish in Cuban waters, enhancing, therefore, existing background knowledge.

Summarizing, the lionfish invasion in the Atlantic Ocean and the Caribbean Sea constitutes a major concern for the scientific community and marine protected area managers, since its colonization capacity and the speed with which it has spread, make it a potential threat to the integrity of invaded ecosystems. This concern has been reflected in the wide range of scientific publications, focused on diverse topics of lionfish in the invaded area, which is notable when compared with those from its native distribution (Fig. 3; Table S1). Lionfish is a new predator with the potential to affect negatively populations of native organisms, by direct predation or by competition for trophic resources. Since the beginning of the invasion, several studies have allowed the evaluation of their impact in different invaded regions. However, contradictory results have been obtained, which generate the current controversy: is lionfish so dangerous, or do the affected ecosystems return to a new state of equilibrium even with the presence of this invader? It is, therefore, necessary to carry out new studies to expand current knowledge about the abundance and distribution of lionfish in invaded ecosystems, its impact on them, the composition of their diet, and the state of the main affected species. New researches should cover broader time scales, enabling a more complete analysis of its impact on native species and ecosystems. The knowledge generated will increase the efficiency of the control and management plans in the invaded areas. These plans should include the development of environmental education programs aimed at the entire population, mainly those living in coastal communities, as well as actions that promote fishing and the consumption of lionfish. New investigations directed to the use of this species, expand the possibilities of management and the implementation of its potential benefits. This should constitute a prioritized line of research. The problem of the invasion continues to exist, and the efforts dedicated to the study of this IAS and the search for solutions should not be diminished.

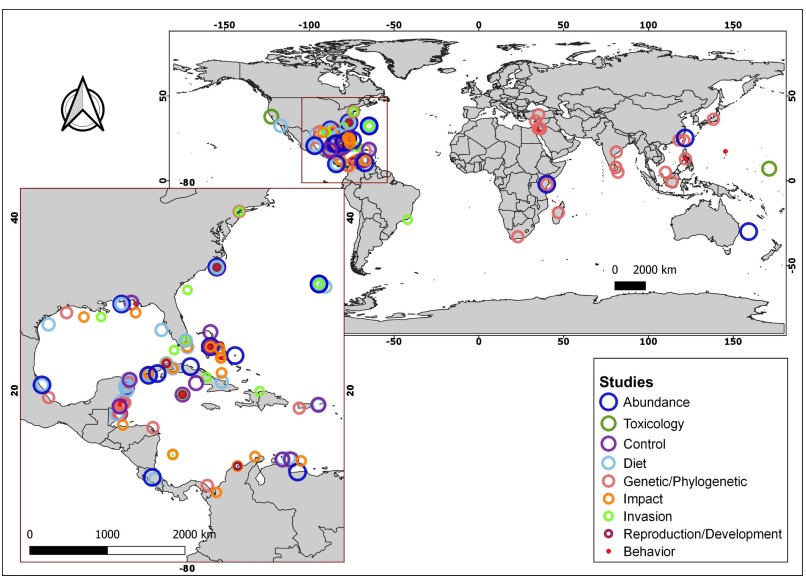

**Figure 3** **Geographical distribution of consulted studies focused on lionfish in specific areas of their native and invaded region categorized by topic.** For additional information about the studies, consult the supplementary material (Table S1).

# ACKNOWLEDGEMENTS

Thanks to Davier Ojeda for their technical assistance in drawing the diagram, to Alejandro Camejo and Miguel A. Ramos for reviewing the English of the manuscript, and also to Yuriem Lezcano for the bibliographic consultancy. We thank the reviewers for their valuable comments on our MS.

## Funding

The authors received no funding for this work.

## Competing Interests

The authors declare there are no competing interests.

## Author Contributions

- Laura del Río performed the experiments, analyzed the data, prepared figures and/or tables, authored or reviewed drafts of the article, and approved the final draft.
- Zenaida María Navarro-Martínez conceived and designed the experiments, performed the experiments, analyzed the data, prepared figures and/or tables, authored or reviewed drafts of the article, and approved the final draft.
- Dorka Cobián-Rojas conceived and designed the experiments, performed the experiments, analyzed the data, authored or reviewed drafts of the article, and approved the final draft.

- Pedro Pablo Chevalier-Monteagudo performed the experiments, analyzed the data, authored or reviewed drafts of the article, and approved the final draft.
- Jorge A. Angulo-Valdes analyzed the data, authored or reviewed drafts of the article, and approved the final draft.
- Leandro Rodriguez-Viera conceived and designed the experiments, performed the experiments, analyzed the data, prepared figures and/or tables, authored or reviewed drafts of the article, and approved the final draft.

## Data Availability

The raw data are available in the Supplemental Files.

## Supplemental Information

Supplemental information for this article can be found online at http://dx.doi.org/10.7717/peerj.15728#supplemental-information.

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
