# Peer review of "Biology and ecology of the lionfish Pterois volitans/Pterois miles as invasive alien species: a review"

_PeerJ, doi:10.7717/peerj.15728_

## Round 0.1 · original submission · Major Revisions

All the reviewers suggested major revisions. Hence my recommendation is to revise the manuscript.

·

Basic reporting

It is a good revision manuscript, the review is of interest and is within the scope of the journal, the information presents the current results and this gives relevance to the review, the structure of the work starting with the Introduction adequately presents the topic and makes clear the work proposal

Experimental design

The structure of the review is adequate and organized, the citations are up to date, the paragraphs are adequate, the subsections could be better organized. The update of the review results is generally good and comments are provided on the revision of the manuscript.

Validity of the findings

The impact of the results are interesting, the review and updating of the literature used for the review is good, the conclusions are also interesting and the research is limited to a review document on the current state of the problem of lionfish, an invader in the great Caribbean. The objectives set are met

Additional comments

It is a good revision work, and although it requires improvements and revisions, it is likely to be published.

Reviewer 2 ·

Basic reporting

-The topic of this review article is within the scope of the journal.
-Professional scientific English usage throughout the manuscript.
-Authors are careless in referencing with some citations unaccounted for. For example "Johnston and Purkis, 2015" and "Chevalier-Monteagudo et al., 2013".
-The Introduction is missing some background. For example, what is the similarity between these two species, Pterois volitans and Pterois miles? Why does authors refer to them interchangeably as "lionfish".

Experimental design

Please clarify whether this review is a systematic literature review. Authors only use google scholar for keyword search, however it is common practise for using scientific databases (ISI WOS/SCOPUS) and Google Scholar next to supplement search.

Validity of the findings

I have no issues with the findings and the figures are presented well. However, whether this manuscript is a systematic literature review still needs to be addressed and formatted accordingly to the methodology.

Annotated reviews are not available for download in order to protect the identity of reviewers who chose to remain anonymous.

·

Basic reporting

The work submitted is clear and concise. There are references missing from other areas of its current distribution. I recommend that information from the Mediterranean Sea is included.

For example:

Sponaugle, S., Gleiber, M. R., Shulzitski, K. and Cowen, R. K. (2019) 'There’s a new kid in town: lionfish invasion of the plankton', Biological Invasions, 21(10), 3013-3018.

Azzurro, E., Stancanelli, B., Di Martino, V. and Bariche, M. (2017) 'Range expansion of the common lionfish Pterois miles (Bennett, 1828) in the Mediterranean Sea: an unwanted new guest for Italian waters ', BioInvasions Records, 6(2), 95-98.

Kletou, D., Hall-Spencer, J. M. and Kleitou, P. (2016) 'A lionfish (Pterois miles) invasion has begun in the Mediterranean Sea', Marine Biodiversity Records, 9(1), 46.

Bariche, M., Kleitou, P., Kalogirou, S. and Bernardi, G. (2017) 'Genetics reveal the identity and origin of the lionfish invasion in the Mediterranean Sea', Scientific Reports, 7(1), 6782.

Poursanidis, D., Kalogirou, S., Azzurro, E., Parravicini, V., Bariche, M. and zu Dohna, H. (2020) 'Habitat suitability, niche unfilling and the potential spread of Pterois miles in the Mediterranean Sea', Marine Pollution Bulletin, 154, 111054.

Experimental design

The design of the study is good but I recommend that information from the Mediterranean Sea is included since its a global review, even if its concentrated for Cuba.

Validity of the findings

Good validity. Please include information from the Mediterranean Sea.

---

## Round 0.2 · accepted · Accept

The authors have addressed the reviewer comments, and I recommend that the manuscript be accepted for publication.

Reviewer 2 ·

Basic reporting

No comment

Experimental design

No comment

Validity of the findings

No comment

Additional comments

The authors have addressed my concerns and I would like to recommend that the manuscript be accepted for publication in its present form.